# Effect of Cell Geometry on the Mechanical Properties of 3D Voronoi Tessellation

**DOI:** 10.3390/jfb13040302

**Published:** 2022-12-16

**Authors:** Zainab Alknery, Zhwan Dilshad Ibrahim Sktani, Ali Arab

**Affiliations:** 1Department of Technical Mechanical and Energy Engineering, Erbil Technical Engineering College, Erbil Polytechnic University, Erbil 44001, Iraq; 2Directorate of Engineering and Projects, Presidency of Salahaddin University-Erbil, Erbil 44001, Iraq; 3Advanced Technology Research Institute, Beijing Institute of Technology, Jinan 250300, China

**Keywords:** additive manufacturing, 3D Voronoi tessellation, bio-inspired, mechanical properties

## Abstract

Irregular 3D biological scaffolds have been widely observed in nature. Therefore, in the current work, new designs are proposed for lightweight 3D scaffolds based on Voronoi tessellation with high porosity. The proposed designs are inspired by nature, which has undoubtedly proven to be the best designer. Thus, the Rhinoceros 7/Grasshopper software was used to design three geometric models for both normal and elongated Voronoi structures: homogeneous, gradient I, and gradient II. Then, stereolithography (SLA) additive manufacturing was utilized to fabricate biopolymeric materials. Finally, a compression test was carried out to study and compare the mechanical properties of the designed samples. The gradient I cylinder show the highest Young’s modulus. For the Homogeneous and gradient II cylinders, elongated Voronoi structures show superior mechanical properties and energy absorption compared to normal Voronoi designs. Hence, these designs are promising topologies for future applications.

## 1. Introduction

Scientists around the world are taking an interest in the recent field of biomedical engineering known as tissue engineering. Tissue engineering is the application of a scientific methodology encompassing biomedical engineering, biomaterials, and transplantation, with the goal of regenerating and growing living tissues or organ substitutes. These alternatives can serve as biological replacements for implants, prostheses [1,2], and scaffolds [3]. A porous scaffold is a crucial component in bone tissue engineering, as it offers momentary structural support for cell development and the formation of new bone tissues. The optimal scaffold is biocompatible, osteoconductive, and conveniently porous, with the correct pore size and excellent 3D interconnectivity. Additionally, the abundance of micro-pores facilitates bone growth [4,5]. The demand for more complex functionality and mechanical stability with good properties is one of the key difficulties confronting tissue engineering today [6].

Moreover, the designer should choose the appropriate scaffold designed with precise specifications, including geometry and size, to achieve the desired properties compatible with natural bone, and this is considered one of the obvious challenges in the current technology. In this regard, both regular and irregular porous scaffolds have gained attention in the literature. Ahamadi et al. [7] investigated the relationship between the relative density and compressive properties for six types of regular scaffolds based on different unit cell configurations (cubic, diamond, truncated cuboctahedron, truncated cube, rhombic dodecahedron, and rhombicuboctahedron). In their study, the geometry of the cell had an obvious influence on the compressive strength and stiffness of 3D-printed samples. Similarly, the work completed by Kantaros et al. [8] clearly showed the dependence of the mechanical properties of scaffolds on the type of unit cell and the unit cell dimensions. Additionally, the 3D truss architecture of regular scaffolds was designed by Shirzad et al. [9], and they optimized their mechanical and physical properties by means of the response surface methodology (RSM). Inevitably, in regular scaffolds, small changes to the unit cell will lead to a global change in the entire structure, and it is difficult to apply local control to the pore shape and the pore size distribution. Thence, irregular scaffold approaches have gained attention to overcome these challenges. 

The irregular porous structure differs from the regular one in several aspects, such as the disarrayed nature of its formation and its nonuniform pore size and pore size distribution in a certain range [10]. The method of designing irregular porous structures based on Voronoi tessellation can be used to design bionic bone trabecular structures. Recently, the development of irregular scaffolds based on Voronoi tessellation has received increasing attention, firstly because this type of scaffold is similar to the human bone microstructure, and secondly because different structural and mechanical performance requirements can be met by adjusting the design parameters. Thirdly, irregular porous structures provide better permeability compared to regular ones, leading to better bone tissue adsorption and regeneration within the scaffold [10,11]. Finally, stress distribution for irregular porous scaffolds is more uniform compared with that in regular ones [12].

Voronoi tessellation is a geometric diagram that reproduces irregular biomimetic cells that can be found in nature [13], such as on turtle shells, giraffes, dragonfly wings, plant leaves, and so on. Because structural topologies have a significant influence on the mechanical behavior of cellular structures, attempts have been devoted to developing deterministic materials with novel mechanical properties [14,15]. Numerous studies have been conducted by researchers using 2D [16,17,18,19] and 3D [20,21,22,23] Voronoi structures. Wu et al. [24] printed a 2D Voronoi structure for a maxillofacial prosthesis to compare the stress distribution in these structures with that in honeycomb, square hole, and round hole structures. They found that during loading, Voronoi structures revealed a more uniform stress distribution, with no separation zone between high- and low-stress areas. Furthermore, Sotomayor and Tippur [25] investigated the effect of both cell irregularity and relative density variation on the mechanical properties of random honeycombs using Voronoi diagrams. An improvement in stiffness of 66% was recorded when the irregular Voronoi structure replaced the regular one. Contrarily, the plastic collapse strength was directly related to the regularity. Nevertheless, with the increase in relative density, a similar collapse strength was observed for irregular and regular structures. These results were further confirmed by Du et al. [26] when they observed a 150% increase in the stiffness when they replaced a regular Voronoi structure with an irregular substructure. In addition, they highlighted the great effectiveness of the Voronoi structure in cases of structural stability with variable load directions and durability with local flaws. Additionally, Bouakba et al. [17] studied the effects of cell geometry and relative density on the stiffness and other mechanical properties of Voronoi-type structures. They proved the superior stiffness of irregular Voronoi structures compared to regular structures. A combination of bending and stretching deformation was found in the cell walls of these structures relying on the relative position of ribs in the assembly. Further evidence accompanying bending and stretching deformations in cellular topologies was supported by the study of Alkhader and Vural [15]. In another work, Zhao et al. [27] designed a gradient scaffold structure based on the Voronoi tessellation method. The results show that a structure with an irregular and gradient design has better performance in terms of mechanical properties. 

Biomimetic concepts, which seek to derive natural inspiration for the development of new engineering and technological solutions, can be used to construct innovative biological scaffolds. A few examples of natural stones that inspired biomedical scaffolds are diabase [28] and basalt [29]. They possess several merits, such as compression strength (350 MPa and 300 MPa, respectively), tensile strength (35 MPa and 30 MPa, respectively), and shear strength (60 MPa). Studies indicated that the strength possessed by these rocks is due to their geometrical structure. Figure 1 shows that these stones are columnar jointed; in other words, they have an elongated shape. This inspired us to implement this design in scaffold applications and evaluate how the elongated Voronoi structure improves the mechanical properties for crashworthiness applications. In the current study, the mechanical properties of the elongated Voronoi structure are compared with those of the normal Voronoi structure in terms of compression strength, stiffness, ultimate tensile strength, and energy absorption. The strengths and weaknesses of both geometries are explained in detail.

## 2. Materials and Methods

### 2.1. Three-Dimensional (3D) Voronoi Structure

In this work, a new bio-inspired geometry was proposed based on 3D Voronoi tessellation. The new design involved replacing the normal Voronoi (N_v_) cell struts with longitudinally elongated Voronoi (E_v_) struts in the Z-direction. The Rhinoceros 7/Grasshopper software was used to design two groups of irregular cylindrical scaffolds: irregular elongated Voronoi structures (IEVSs) and irregular normal Voronoi structures (INVSs). Each group contained three models, as shown in Figure 2:I.Homogeneous cylinder E1 and N1: the number of seeds was evenly distributed throughout the cylinder with the dimensions R = 3.27 mm and H = 12 mm, as shown in Figure 3. The number of distributed seeds was 110 and 200 for E1 and N1, respectively.II.Gradient I cylinder E2 and N2: the cylinder was divided into two regions; the middle was hollow, so the core of the cylinder did not have any cells. The remainder was the perimeter, which had several seeds distributed throughout it, numbering 125 and 250 for E2 and N2, respectively, with the dimensions R = 3.27 mm, r = 1 mm, and H = 12 mm.III.Gradient II cylinder E3 and N3: the same as the previous design, but with a difference in that the core was filled with seeds. The core had approximately 20 seeds while the perimeter had 100 and 215 seeds for E3 and N3, respectively. Hence, the core was less dense compared to the perimeter.

The process, as indicated in Figure 4, began by distributing several points randomly in the 3D cylindrical normal Voronoi structure to obtain a specific volume with defined dimensions. Then, the scale command was applied to create the 3D cylindrical elongated Voronoi structure. The porosity was determined by the gravimetric method, as shown in Equations (1) and (2) [30,31,32,33].
(1)$$\rho_{\mathrm{scaffold}}=\;\mathrm{mass}/\mathrm{volume}$$
(2)$$\mathrm{Total}\;\mathrm{porosity}=1-(\rho_{\mathrm{scaffold}}/\rho_{\mathrm{material}})$$
where $\rho_{\mathrm{material}}$ is the density of the material and $\rho_{\mathrm{scaffold}}$ is the apparent density of the scaffold.

### 2.2. Printing

A commercial plant-based resin (made in China, Shenzhen, Guangdong, ANYCUBIC 3D Technology Company) was used; its properties are given in Table 1. Rhinoceros 7/Grasshopper was used to prepare the Voronoi tessellation of all 3D models. The CAD model was used to refer to the designed model. Following the design steps, six samples were saved as an STL file before being sent for printing. The details of the construction steps performed by the Grasshopper software are explained in Appendix A. The SLA technique was used to print the samples due to its precision in printing geometric shapes with complex details. The SLA printer contains a build platform, which is the place where the part is created. Underneath, there is a resin tank, and clear glass allows the UV laser to cure the resin. In order to begin printing, the model file must first be uploaded, and then the tank must be filled with resin up to the limited level. The laser passes back and forth inside, eventually solidifying the liquid plastic. Finally, the printed part is taken out to be washed in rubbing alcohol to remove the excess resin. Figure 5 shows the ELFIN 3 Mini by NOVA3D Printer, which was used to print the test samples for this work. Parameters were identified and controlled by the NOVA Printing control software as follows: layer thickness of 0.06 mm, printing speed of 50 mm/h, 119.8× 67.8× 149.9 mm printing size, and a 90° printing angle with ±0.1 mm model tolerance.

### 2.3. Mechanical Testing

#### 2.3.1. Compressive Testing

The uniaxial compression tests were conducted on six lattice models with low-density specimens using a GOTECH AI-3000 universal testing machine with a total height of 1570 mm. The speed was kept constant at 5 mm/min. The test was performed in a quasi-static condition at a strain rate of 0.001/s. The applied load and displacement were measured by a digital board, and compression tests were conducted at least four times for each sample. Then, the stress–strain curve was plotted by taking the average of several curves. The apparent stress and apparent strain can be calculated from Equations (3) and (4), respectively.
(3)$$\sigma^{*}=F/A=F/\pi \;R^{2}$$
(4)$$\varepsilon^{*}=\Delta L/L_{i}=\frac{\left(L_{i}-L_{f}\right)}{L_{i}}$$
where *R* is the radius of the scaffolds in mm, $L_{i}$ is the initial length of the porous scaffold in mm, $L_{f}$ is the length of the scaffold after the deformation in mm, and ΔL represents the compressive displacement (mm), based on the concept that the apparent stress is constant over the cross-sectional area and throughout the gauge length.

#### 2.3.2. Stress–Strain Curve

The apparent stress and apparent strain are obtained by applying a load to a test specimen progressively and monitoring the resulting deformation. In addition, the form of this deformation can be compression, torsion, or stretching. Numerous properties of a material are revealed through this curve, including the apparent Young’s modulus E*, stiffness S*, maximum deformation $w_{max.}$, and ultimate strength US. There are two ways to draw this curve: the conventional stress–strain diagram and the true stress–strain diagram [35]. 

In the current study, the conventional method was used, in which the engineering stress can be calculated by dividing the applied load by the cross-sectional area of the sample (see Equation (3)). In order to compare the mechanical properties of the models, the apparent Young’s modulus and stiffness were calculated from Equations (5) and (6), respectively, while the ultimate strength could be determined precisely from the apparent stress–strain curve.
(5)$$E^{*}={\;\sigma }^{*}/\varepsilon^{*}$$
(6)$$S^{*}=\;F/\Delta L$$

#### 2.3.3. Total Energy Absorption (TEA)

Energy absorption is a key parameter in biomedical engineering, especially for scaffold design in the event of a bone crash. Because of the difference in geometrical designs between IEVSs and INVSs, energy absorption was determined to compare their abilities to absorb energy. The TEA is obtained from the area under the load–displacement curve using Equation (7) [36]:(7)$$\mathrm{TEA}=\int P_{\mathrm{ave}}{\;d}_{s}\;\equiv {\;P}_{\mathrm{ave}}\;\left(d_{f}-{\;d}_{i}\right)$$
where $P_{\mathrm{ave}}$ is the mean crushing force, $d_{i}$ is the initial crushing distance, and $d_{f}$ is the final crushing distance. Commonly, TEA is measured with kJ according to the SI unit.

#### 2.3.4. Specific Energy Absorption (SEA)

Since our proposed models were designed to be used in the manufacturing of biological scaffolds, they required a light weight. Therefore, it was necessary to calculate the SEA of the models. SEA is defined as the total energy absorbed divided by the mass of the sample, with the unit of kJ/kg [37]. Consequently, this can be determined based on Equation (8):(8)SEA= TEA/MASS

## 3. Results and Discussion

### 3.1. Results

#### 3.1.1. Density and Porosity of Samples

The porosity not only affects the mechanical properties but can also have a direct effect on bone growth in the bone scaffold. For the bone scaffold, it was suggested that the porosity should be over 60% and the pore size should be greater than 300 µm, as this geometry could promote bone formation [38,39,40]. By using Equations (1) and (2), the porosity and density of the design were calculated. The volume of the structures was controlled by the number of Voronoi cells, and thus the density distribution of the scaffold structure was controlled. All models were kept at the same volume of approximately 95 ${\mathrm{mm}}^{3}$ and 0.0005 $g/{\mathrm{mm}}^{3}$ density with a thickness of 0.1 mm. The microstructures of bone are diverse. The porosity was set at around 60% for all samples, as indicated in Table 2.

#### 3.1.2. Mechanical Properties

Sufficient compressive strength is a primary and prominent requirement for bone scaffolds in orthopedic applications. Hence, a compressive strength test was essential for evaluating the mechanical properties of different Voronoi designs. The compressive test was repeated four times for each sample, and the average value of these four tests is reported here. Figure 6 illustrates the load–displacement curves of the Voronoi structures with different designs. The stiffness of the structures can be determined by the load–displacement curve, which shows that the normal Voronoi structures are stiffer. Figure 6a reveals that both E1 and N1 broke at similar displacements: E1 broke at 4.0 mm with a stiffness of 352.5 ± 0.98 N/mm, and N1 broke at around 3.8 mm with a stiffness of 380.3 ± 1.02 N/mm. Similarly, as shown in Figure 6b, E2 and N2 fractured at similar approximate displacements: E2 was broken at 2.0 mm displacement with a stiffness of around 286.9 ± 1.06 N/mm and N2 at 1.8 mm displacement with 272.8 ± 1.04 N/mm stiffness. It is noteworthy that the gradient I Voronoi structure for both E2 and N2 shows the lowest stiffness among the three designs. However, the gradient II Voronoi structures (E3 and N3) showed a different trend, as shown in Figure 6c. E3 recorded a maximum displacement of 3.6 mm, which is the highest among all samples, with a stiffness of 302.8 ± 0.95 N/mm, while N3 recorded 2.1 mm of displacement with a stiffness of 336 ± 0.94 N/mm.

Table 3 shows the mechanical properties of the various designs. It is clear that the elongated Voronoi structure gradient I type (E2) has the highest apparent Young’s modulus of 142 ± 0.66 MPa, while the greatest ultimate strength is achieved by the elongated Voronoi structure homogeneous type (E1), at 29.00 ± 0.12 MPa. Moreover, the normal Voronoi structure homogeneous type (N1) shows a maximum stiffness of 380.30 ± 1.02 N/mm. It should be noted that there was not a significant difference between the two types of design (elongated and normal Voronoi) in terms of maximum deformation. The ratios are either very close or exactly equal, as in E1 and N1 and E2 and N2, respectively, except for the designs E3 and N3, in which there is a colossal difference, where the elongated Voronoi structure is 43% superior to the normal Voronoi structure.

In addition, Figure 7 displays the apparent stress–strain curves of all the 3D Voronoi structures (E1 and N1, E2 and N2, and E3 and N3). It is evident from the data that the apparent strain increases with increasing load, and all designs exhibited a linear increase in apparent stress up to a specific apparent strain value, after which it shifted to a more moderate relationship between these two parameters. Figure 7a clarifies the apparent stress–strain comparison for both homogeneous designs (elongated and normal Voronoi structures). Both designs display a linear relationship between stress and stress parameters until they reach the same apparent strain of 0.1. Nevertheless, E1 has a higher apparent stress (19.2 MPa) than N1 (18.8 MPa). The apparent stress then begins to rise by gradually increasing the apparent strain until the samples reach the maximum deformation and are fractured at an apparent strain of 0.3 for both E1 and N1, with apparent stresses of 28.2 MPa and 27.0 MPa, respectively. Although E1 appears to withstand more apparent stress than N1, the difference is very small, and its effect is negligible when used in applications that are based on pressure. Unlike the homogeneous design, the gradient Voronoi structures show a noticeable difference in withstanding the applied load. Figure 7b illustrates that the maximum apparent stress of E2 (16.1 MPa) outperformed that for N2 (15.3 MPa). However, they broke at a lower apparent strain (around 0.1). This indicates that, despite the fact that the gradient I E2 begins to deform at higher apparent stresses than the gradient I N2, they do not withstand the high apparent stresses achieved by the homogenous designs (E1 and N1). Furthermore, their maximum apparent stress is also lower than that of the gradient II Voronoi structures (E3 and N3), which in turn is lower than that of the homogeneous Voronoi structures, as shown in Figure 7c. In addition, the maximum deformation reached 0.33 for E1 and 0.31 for N1, respectively, which is higher than that of gradient I Voronoi structures and somewhat similar to that for the homogeneous Voronoi structures.

#### 3.1.3. Energy–Displacement Curve

Figure 8 shows the comparison between the energy absorption and displacement for 3D Voronoi structures (E1 and N1, E2 and N2, and E3 and N3, respectively). It is noticeable that the relationship between energy and displacement demonstrates an increasing trend for both structures (elongated and normal Voronoi). From Figure 8a, it is clear that for the homogeneous structures (E1 and N1), the rate of increase in displacement up to 1.0 mm results in a minor rise in energy absorption. However, a further rise in displacement (more than the mentioned rate) results in a ramping up of energy absorption, reaching a plateau at 4.0 mm. This result reveals that further displacement is beneficial for both structures to improve their energy absorption, and it indicates that the two designs show similar behavior in terms of energy absorption improvement. Meanwhile, Figure 8b presents the comparison between the energy absorption and displacement for the gradient I E2 and N2 structures. Similarly, the increment in the displacement up to a certain value (0.8 mm) results in a moderate improvement in energy absorption for both models. Nonetheless, when these samples are subjected to a higher rate of displacement, it causes an abrupt enhancement in energy absorption, reaching the maximum (0.55 J) at a displacement of 2.0 mm for E2, and 0.37 J at a displacement of 2.0 mm for N1. Meanwhile, the gradient II structures E3 and N3 show better performance in terms of energy absorption than the previous design (gradient I). It can be seen from Figure 8c that the greatest energy absorption was obtained at 4.0 mm displacement with approximately 1.79 J for E3, and displacement of 2.2 mm with energy absorption of 0.80 J for N1.

On the other hand, the SEA for all six designs is illustrated in Figure 9. It is noticeable that, comparing the two types of design, the elongated Voronoi structures achieved a higher rate of specific energy absorption, as they outperformed the normal Voronoi structures with the same relative density at a constant crushing velocity. The homogeneous design (E1) features have the best performance among the six Voronoi topologies, with specific energy absorption of 52.45 J/g, followed by N1, with specific energy absorption of 45.74 J/g. On the contrary, the lowest performance is provided by the gradient I N2, with specific energy absorption of approximately 6.27 J/g.

### 3.2. Discussion

#### 3.2.1. Mechanical Properties and Energy Absorption

In this research, the mechanical properties of different Voronoi structure designs were examined. The experimental results indicate that changes in the cell geometry of Voronoi structures could lead to completely different mechanical performance. The Voronoi structure is attractive for bone scaffold design, as its irregularity is biomimetic of bone trabeculae and, as mentioned above, it can be optimized to display in-demand mechanical properties. It is noteworthy that, despite the similar displacement trend of both designs, the elongated Voronoi structure resists fracture at a higher displacement than the normal Voronoi structure. Hence, considerable energy absorption is presented when the elongated Voronoi structures replace the normal Voronoi structures. Experimental results show that the elongated Voronoi structures have better mechanical performance. Nevertheless, more stretching of the cells will show low levels of anisotropy [41].

On the other hand, in terms of comparison between the design types for manufactured models, the gradient I Voronoi structure model for both elongated and normal designs (E2 and N2) presented the highest apparent Young’s modulus among all models. This is due to the microstructure and design features, where all density gathers at the edges, providing optimum structural support for the scaffold [42], as well as making it biocompatible with natural bone [43]. Meanwhile, the reason that the gradient II Voronoi structure (E3 and N3) shows a lower apparent Young’s modulus is due to the structural design, as well as to the fact that it has higher porosity, which is necessary for the scaffold in order to enable the rapid absorption of materials, which causes it to break at lower apparent stress and means that it does not resist the imposed load for a long time. Whereas scaffolds with high porosity are typically weaker because the pores cause Voronoi cells to be separated by a greater distance, scaffolds with low porosity are stronger. Nevertheless, it is stiffer than the gradient I Voronoi structure due to the improved geometric design, which was confirmed by finite element analysis (FEA) [44,45]. Whilst the homogeneous design of both elongated and normal Voronoi (E1 and N1) exhibits more stiffness and ultimate strength than the rest, this is due to the fact that when homogeneous structures are subjected to a load, this load will be distributed largely evenly over all parts of the structure, and thus the structure absorbs the largest amount of energy. In addition, homogeneous structures are widely preferred in fabricating scaffolds, as they can provide a biological environment that improves cell proliferation [46,47].

To further our comprehension of the mechanical response of the proposed Voronoi structures, the deformation patterns of the elongated and normal Voronoi structures are shown in Figure 10. The deformation of both designs shows a Y-shaped joint and an X-shaped joint. When the material was compacted, these joints revolved around the cross-center throughout the compression process until they were crushed. Ordinarily, the rotations of the X-shaped joint could absorb more energy than the rotations of the Y-shaped joint. It is important to mention that there are more X-shaped joints in elongated Voronoi structures than in normal Voronoi structures. This might be the reason that the elongated Voronoi structure has a higher energy absorption capacity than the normal Voronoi structure [48]. Du et al. [49] reported that stress was mainly concentrated at the joints connected by the struts in a cubic scaffold. Studies in the literature reveal that in most scaffold structures, stress is severely concentrated at the sharp edges, and a normal Voronoi structure has a higher number of sharp edges compared to an elongated Voronoi, which can lead to earlier failure in normal Voronoi structures [45].

#### 3.2.2. Failure Mode

The failure mode is the same for all samples, as illustrated in Figure 11, for a homogeneous elongated Voronoi structure. The cylindrical Voronoi cells all eventually collapsed due to a failure mode involving three subsequent stages of failure, starting from the fracture of the cells’ strut, cell collapse, and, finally, fracture growth [50]. This demonstrates that despite the material’s great strength, the samples experienced a fracture and so absorbed less energy. A similar phenomenon of failure was detected in the breaking behavior of lattice structures for diverse additively manufactured materials using the same technique [51,52]. Moreover, Chao et al. [53] studied the failure nature of a Voronoi structure with different porosity levels (70, 80, and 90%) using the finite element method (FEM). They found that the maximum Von Mises stress in porous Voronoi structures is mainly concentrated at the nodes where the connecting rods of the porous structure are connected. They observed that the randomly distributed pores were helpful to promote fragile and brittle pore edges, which led to the concentration of more stresses. The most obvious finding in their study is that the average pore size has a crucial influence on the stress distribution on the connecting rod edges. The smaller the pore size, the higher the stress concentration, which leads to easier fracture. They suggested that when designing the Voronoi structure, the appropriate average cell size should be considered more than the porosity percentage. Based on the results presented in their study, combined with the fact that the elongated Voronoi structures in the current study have larger pore sizes, they can carry larger apparent stresses and loading before they start to be fractured. This is a reason for the mechanical properties’ improvement in the current study. In addition, no buckling behavior was observed in the structures because the Voronoi cells consist of inclined struts, which provide support for the structure to tolerate the buckling load [54]. Thus, they are desirable for load-bearing applications because they provide stability and easy predictability of the mechanical response [55].

## 4. Conclusions

The aim of this work was to improve the mechanical properties of 3D biological scaffolds by presenting a new design based on changing the geometric structure of the normal Voronoi structure by changing the direction of the cell struts and stretching them in the Z-direction. The proposed models were designed with the Rhinoceros 7/Grasshopper software and then 3D-printed using the SLA technique. The major conclusions are formulated as follows:IEVSs were proven to bear higher stress than INVSs, as well as outperforming them in terms of energy absorption.IEVSs showed better performance in terms of resistance to fracture, with a higher displacement rate than INVSs, which explains their high energy absorption.In terms of the three designed models, the gradient I Voronoi structure model (E2 and N2) presented a higher apparent Young’s modulus than the gradient II Voronoi structures and the homogeneous structure due to the improved design features. Meanwhile, the homogeneous structures exhibited greater stiffness than the rest.In general, the mechanical properties are greatly affected by the geometric design of the Voronoi structures. For the gradient I and II cylinders, elongated Voronoi structures possess superior mechanical properties compared to normal Voronoi structures.

## Figures and Tables

**Figure 1 jfb-13-00302-f001:**
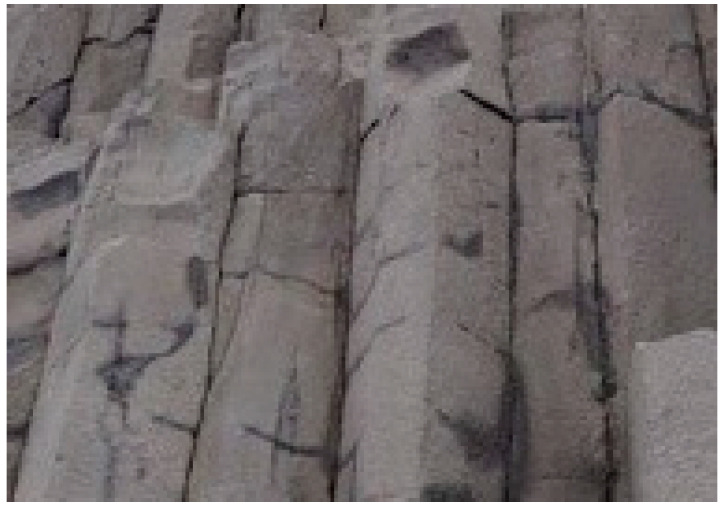
Columnar jointed rocks. “Reprinted/adapted with permission from Ref. [28]. 2022, 2022 Rockhound Resource”.

**Figure 2 jfb-13-00302-f002:**
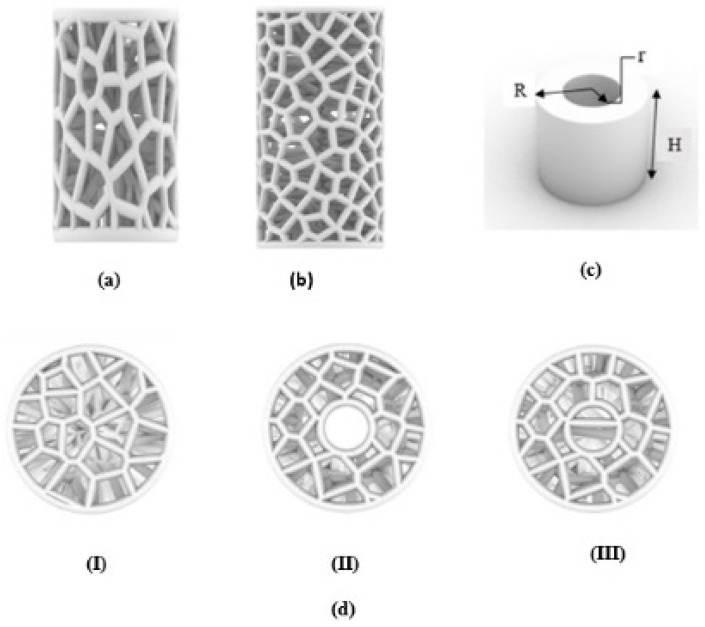
An illustration of designed irregular scaffolds: (**a**) elongated Voronoi; (**b**) normal Voronoi; (**c**) cylinder details; (**d**) the top view of the three models.

**Figure 3 jfb-13-00302-f003:**
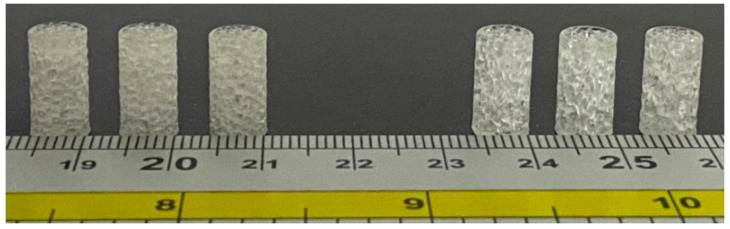
Dimensions of the fabricated samples.

**Figure 4 jfb-13-00302-f004:**
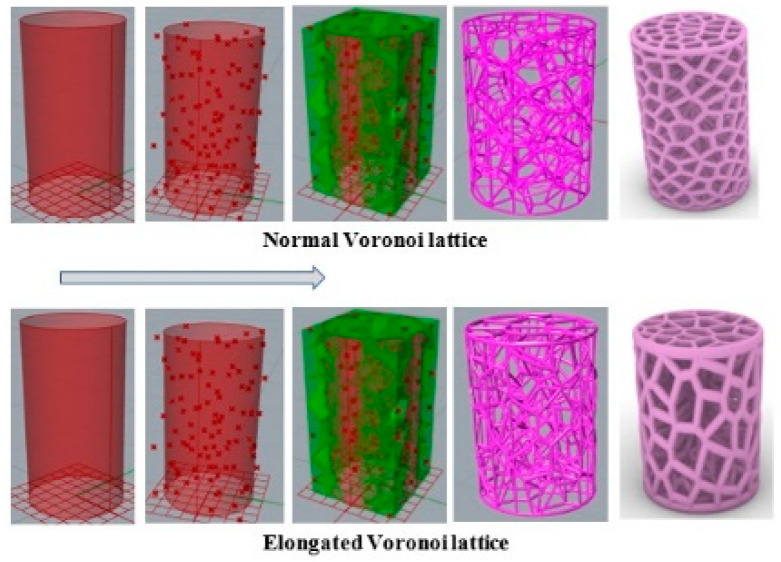
Steps involved in designing the models in Grasshopper software (Rhinoceros version 7.17.22102, Robert McNeel & Associates company, North America, and Pacific).

**Figure 5 jfb-13-00302-f005:**
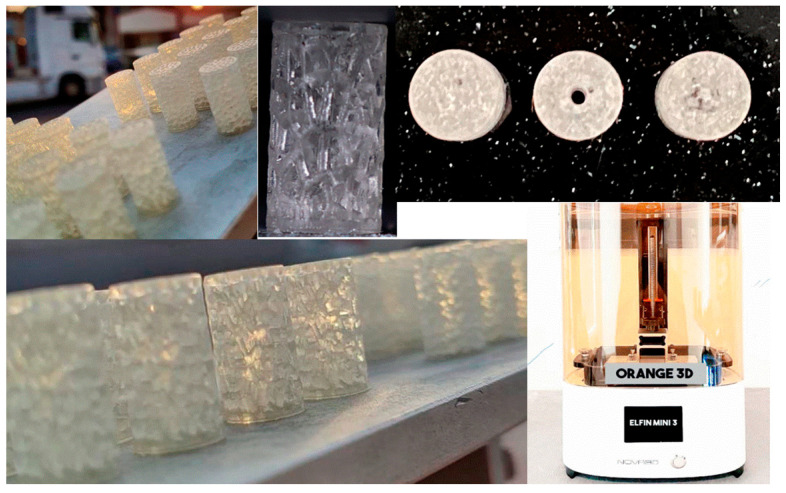
NOVA 3D printer and the printed samples.

**Figure 6 jfb-13-00302-f006:**
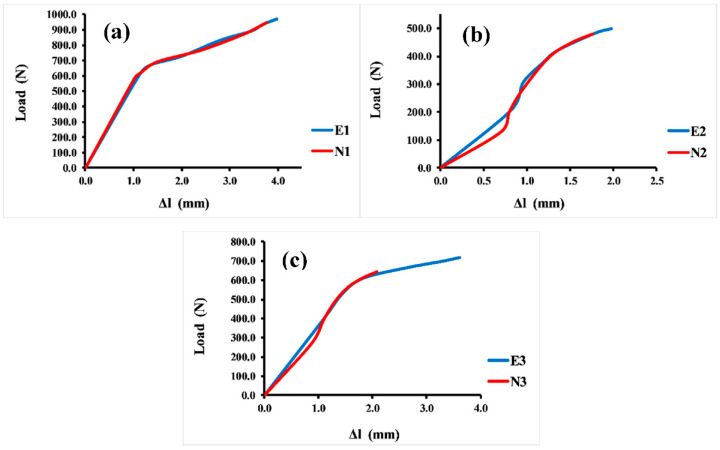
Load–displacement curve for the Voronoi structures. (**a**) Homogeneous model for elongated (E1) and normal (N1) Voronoi structures; (**b**) gradient I model for elongated (E2) and normal (N2) Voronoi structures; (**c**) gradient II model for elongated (E3) and normal (N3) Voronoi structures.

**Figure 7 jfb-13-00302-f007:**
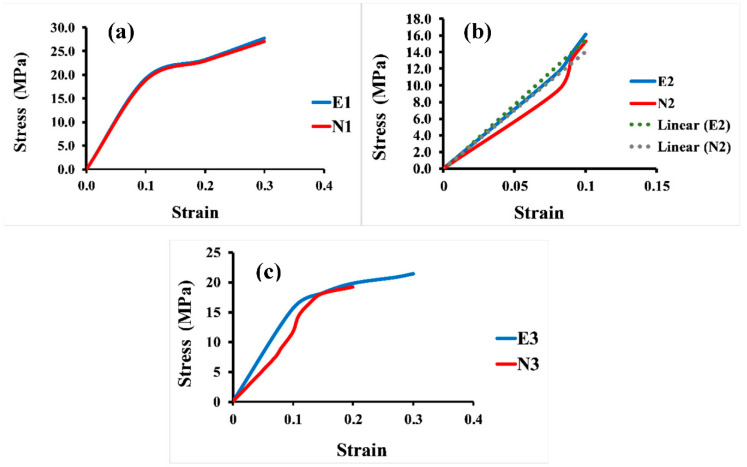
Stress–strain curve for the six samples under uniaxial compression load. (**a**) Homogeneous model for elongated (E1) and normal (N1) Voronoi structures; (**b**) gradient I model for elongated (E2) and normal (N2) Voronoi structures; (**c**) gradient II model for elongated (E3) and normal (N3) Voronoi structures.

**Figure 8 jfb-13-00302-f008:**
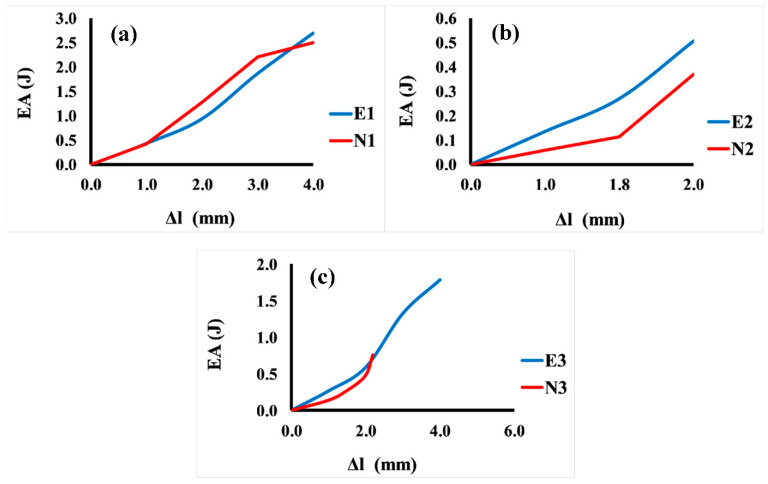
Variation in energy absorption (EA) as a function of displacement for the six Voronoi structures. (**a**) Homogeneous model for elongated (E1) and normal (N1) Voronoi structures; (**b**) gradient I model for elongated (E2) and normal (N2) Voronoi structures; (**c**) gradient II model for elongated (E3) and normal (N3) Voronoi structures.

**Figure 9 jfb-13-00302-f009:**
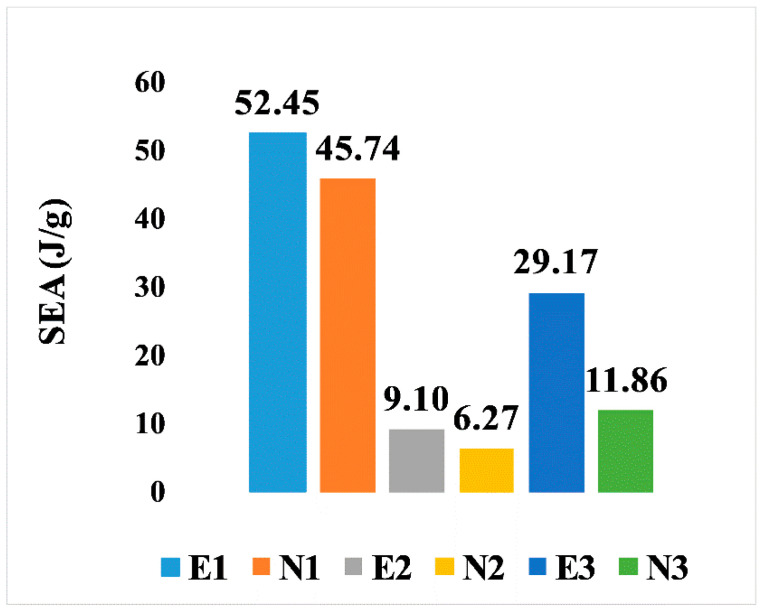
Specific energy absorption of the six models.

**Figure 10 jfb-13-00302-f010:**
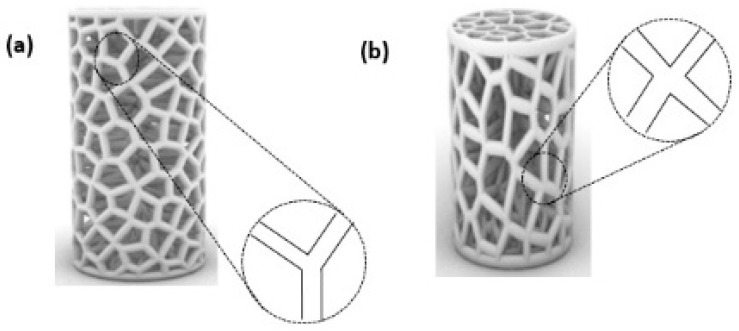
Deformation pattern of the (**a**) normal Voronoi structures; (**b**) elongated Voronoi structures.

**Figure 11 jfb-13-00302-f011:**
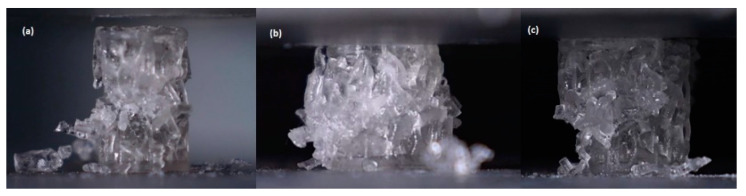
Failure mode of homogeneous elongated Voronoi structure. (**a**) Fracture prediction starts; (**b**) collapse; (**c**) fracture propagation.

**Table 1 jfb-13-00302-t001:** Biopolymer/plant-based resin parameters [34].

Viscosity	Shrinkage	Vitrification Temp	Extension Strength	Bending Strength	Elongated Break
(MPa·s) (25 °C)	(%)	(°C)	(MPa)	(MPa)	(%)
150–350	1.88–2.45	60∓5	35–45	40–50	8–12

**Table 2 jfb-13-00302-t002:** Geometry properties of the Voronoi structures.

Name of Design	Unit	E1	E2	E3	N1	N2	N3
Mass	g	0.20	0.20	0.20	0.20	0.20	0.20
Porosity	%	64 (±1.87)	60 (±1.23)	64 (±0.92)	64 (±0.84)	60 (±1.02)	64 (±0.78)

**Table 3 jfb-13-00302-t003:** Mechanical properties of the six Voronoi structures.

Mechanical Properties	Unit	E1	E2	E3	N1	N2	N3
E*	MPa	128 (±0.64)	142 (±0.66)	115 (±0.59)	120 (±0.65)	121 (±0.62)	112 (±0.61)
S*	N/mm	352.51 (±0.98)	286.90 (±1.06)	302.80 (±0.95)	380.30 (±1.02)	272.80 (±1.04)	336 (±0.94)
US	MPa	29.00 (±0.12)	16.30 (±0.11)	21.39 (±0.16)	28.22 (±0.13)	15.80 (±0.12)	19.18 (±0.18)
$w_{max.}$	mm	0.33 (±0.02)	0.10 (±0.01)	0.30 (±0.04)	0.31 (±0.02)	0.10 (±0.01)	0.17 (±0.04)

## Data Availability

Not applicable.

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
