# Peer review of "Effect of Cell Geometry on the Mechanical Properties of 3D Voronoi Tessellation"

_jfb, 2022, doi:10.3390/jfb13040302_

Round 1

Reviewer 1 Report

The manuscript discussed the Effect of cell geometry on the mechanical properties of 3D Voronoi tessellation. The manuscript can be published after minor revision.

1.      Figure 1 is provided twice. Remove repetition.

2.      Failure mode of Voronoi structures needs to be explained.

3.      Table 4, if values are taken from another study, provide its references.

4.      Quality of Figure 6 & 7 need to be improved.

Author Response

Dear Reviewer,

Thank you very much for your time involved in reviewing the manuscript and your very encouraging comments on the merits.

Comments and Suggestions for Authors

The manuscript discussed the Effect of cell geometry on the mechanical properties of 3D Voronoi tessellation. The manuscript can be published after minor revision.

  1. Figure 1 is provided twice. Remove repetition.

Response: Thank you for your professional advice. We removed the repeated Figure 1.

  1. Failure mode of Voronoi structures needs to be explained.

    Response: Thank you very much for your detailed review. Here this section was modified as following:

The failure mode is the same for all samples as illustrated in Error! Reference source not found. for homogeneous elongated Voronoi structure. The failure mode is the same for all samples as illustrated in Error! Reference source not found. for homogeneous elongated Voronoi structure. The cylindrical Voronoi cells all eventually collapsed due to a failure mode involving three subsequent stages of failure starting from the fracture of the cells’ strut, cell collapse, and finally, it followed by fracture growth [31]. This demonstrates that despite the material's great strength, the samples experienced a fracture and so absorbed less energy. This is due to cracking only occurring at low strains. Notably, the densification of the structures happened at varying strains of about: 22% for homogeneous, 9% for gradient I, and 17% for gradient II structures, until the entire structure was flattened. This is made abundantly clear by the gradual rise in the strain that occurs as a result of the pores closing and the struts coming into contact with each other under higher strain [55]. A similar phenomenon of the failure was detected in the breaking behaviour of lattice structures for diverse additive manufactured materials using the same technique [56,57]. Moreover, Chao et al [58] Studied the failure nature of the Voronoi structure with different porosity levels (70, 80, and 90%.) using the Finite Element Method (FEM). They found that the maximum Mises stress (from Von Mises Yield criterion theory) from porous Voronoi structures is mainly concentrated at the nodes where the connecting rods of the porous structure are connected. They investigated that the randomly distributed pores are helpful to promote fragile and brittle pore edges which led to the concertation of more stresses. The most obvious finding in their study is the average pore size has a crucial influence on the stress distribution on the connecting rod edges. The smaller the pore size, the higher the stress concentration which leads to easier fracture. They suggested that for designing the Voronoi structure, the appropriate average cell size should be considered more than the porosity percentage. Based on the results presented in their study and combining the fact that the elongated Voronoi structures in the current study have larger pore sizes; they can carry larger apparent stresses and loading before they start to be fractured. This is a reason for the mechanical properties’ improvement in the current study.

  1. Table 4, if values are taken from another study, provide its references.

Response: Thank you very much for notices. We have removed Table 4.

  1. Quality of Figure 6 & 7 need to be improved.

    Response: Thank you very much for your thorough observation. Your advice is quite supportive for improving our manuscript. The quality of Figure 6 and Figure 7 are improved.

Reviewer 2 Report

The manuscript shows details a novel topology, referred as Elongated Voronoi structures and compares their behavior to “normal” Voronoi Structures. The authors produce variations of these models by SLA and test them their mechanical behavior by uniaxial compression. Conclusions are drawn to compare their properties and establish the benefits of this new model. There are, however, numerous shortcomings that compromise the quality of the manuscript:

(i) The authors should carefully prepare the manuscript before submission. There numerous issues that must be addressed that are still related with the preparation stage of the manuscript (e.g. GARGHICAL BSTRACT, instead of graphical abstract).

(ii) Abstract – “Hence, it is preferred for scaffold designs.” – Maybe “Hence, these design are promising topologies for future applications”

(iii) “Researchers and designers are always looking for ways to reduce the weight of structures while maintaining their structural integrity and mechanical properties” – Do the authors mean specific mechanical properties (i.e. mechanical properties normalized by their mass)?

(iv) Figure 1 is repeated. Are these images property of the authors? If not, they should be referenced.

(v) “Then the stress-strain curve was plotted according to Hooke’s law” – this does not make sense for non-linear behaviors, especially at higher strains.

(vi) Eqs. 3 and 4 should be Apparent stress and strains (σ* and ε*) since they refer to the lattice, not the base material. Also, the “r” in Eq.3 is not the same as the “r” in Figure 2.

(vii) “Young’s modulus and 193 stiffness were calculated from the Eq. (5) and Eq. (6)” – the modulus that is calculated in this test is the apparent modulus (E*). It is the elastic modulus of a lattice, not a bulk material.

(viii) “Energy Absorption (EA)” – sometimes the authors use EA, other times they use TEA

(ix) “?Ì… is the relative density of the samples of about 0.36” – This is incoherent considering that (x) Table 2 shows a ~60-64% porosity. Also, standard deviations should be added.

(xi)On the other hand, when the load surpasses the Euler buckling load, the cell wall of the Voronoi structures buckles, this is called the elastic collapse, which is similar to that in foam and honeycomb.” Considering that the authors statement is located in the methods section, they cannot yet prove this. In fact, even for honeycombs, this is only true for one direction. Also, honeycomb theory (especially, when citing L.Gibson and M.Ashbys work) is appropriate for high values of porosity (>90%). Considering that the samples display a porosity ~60-64% (Table 2), there are strong axial and hinging effects that affect the micromechanical deformation behavior of the samples.

(xii) “The compressive test was repeated 4 times for each sample and the average value of these four tests was reported here” – Standard deviations should be added.

(xiii) Lines 171 to 313 – Should be corrected to Apparent stresses, strain and moduli.

(xiv) Table 3 – add standard deviations and keep the same number of decimals.

(xv) Displacements – sometimes in “m”, others in “mm”. The same unit should be used.

(xvi) Fig.9 - why are so many decimals used? Keep the coherence with other tables and figures.

(xvii) Table 4 – Where the C1 and C2 values determined by testing only one topology. Usually, these factors are determined by testing the same topology with different specific densities (e.g. by changing the rib thickness) following by non-linear regression. Considering that this model is based on an exponential function, how can the authors claim that these parameters may be determined with only one result?

(xviii) “Thus it can be concluded from this that gradient I structure design can resist bending more than the fully filled structure and therefore its value is expected to exceed 1 in the case of using a metallic material.” – considering that these models are based on normalized moduli (E*/Es), this is simply not true. Surely, the changes in C1 (admitting that these values are correct, even though they were calculated with only one point per topology) reveal changes in the deformation mechanism, however, they are not correctly interpreted.

(xix) "The stress-strain curve shows brittle behavior and this was expected because it is known that most polymeric materials are brittle” – This is not true, and the apparent stress-strain plots prove this. Also Reference 55 detail PLA.

(xx) Samples are called E1-E3 and N1-N3, however, in the conclusions section are referred as IEVs and INVs.

(xxi) Final remark, considering that Voronoi structures are usually generated randomly, what is the impact of this aspect in this study?

Author Response

Dear Reviewer,

Thank you very much for your time involved in reviewing the manuscript and your very encouraging comments on the merits.

Comments and Suggestions for Authors

The manuscript shows details a novel topology, referred as Elongated Voronoi structures and compares their behavior to “normal” Voronoi Structures. The authors produce variations of these models by SLA and test them their mechanical behavior by uniaxial compression. Conclusions are drawn to compare their properties and establish the benefits of this new model. There are, however, numerous shortcomings that compromise the quality of the manuscript:

Response: Thank you for your professional advice. We will consider your valuable recommendations. They are helpful to improve the quality of the current manuscript to make it satisfactory for the publication in high reputed journals.

(i) The authors should carefully prepare the manuscript before submission. There numerous issues that must be addressed that are still related with the preparation stage of the manuscript (e.g. GARGHICAL BSTRACT, instead of graphical abstract).

Response: Thank you for your professional advice. We corrected this spelling mistake.

(ii) Abstract – “Hence, it is preferred for scaffold designs.” – Maybe “Hence, these design are promising topologies for future applications”

Response: Thank you for your professional advice. Your valuable advice is very useful. That is the reason we considered it to make the manuscript objective more professional. We have rewritten the last sentence in the abstract as following:

Hence, these designs are promising topologies for future applications

(iii) “Researchers and designers are always looking for ways to reduce the weight of structures while maintaining their structural integrity and mechanical properties” – Do the authors mean specific mechanical properties (i.e. mechanical properties normalized by their mass)?

 Response: Thank you very much for your thorough observation. We mean mechanical properties such as Young’s modulus, stiffness, and ultimate strength.

(iv) Figure 1 is repeated. Are these images property of the authors? If not, they should be referenced.

Response: We really appreciate your suggestions. We removed the repeated Figure 1. We cited the reference Figure 1.

(v) “Then the stress-strain curve was plotted according to Hooke’s law” – this does not make sense for non-linear behaviors, especially at higher strains.

Response: Thank you very much for your valuable suggestion. Based on your scientific advice, we redrew the stress and strain curve. Kindly find Figure 6 and Figure 7. Also, the new corresponding values and explanation of mechanical behaviour were added in the text and highlighted.

(vi) Eqs. 3 and 4 should be Apparent stress and strains (σ* and ε*) since they refer to the lattice, not the base material. Also, the “r” in Eq.3 is not the same as the “r” in Figure 2.

Response: Thank you very much. You are right, stress and strain are referred to the lattice, and in new version we replaced stress and strain by the (σ* and ε*), and in the equation 3 we replaced r with R.  

(vii) “Young’s modulus and 193 stiffness were calculated from the Eq. (5) and Eq. (6)” – the modulus that is calculated in this test is the apparent modulus (E*). It is the elastic modulus of a lattice, not a bulk material.

Response: Thank you very much for your thorough observation. E is change to the E* in equation (5).

(viii) “Energy Absorption (EA)” – sometimes the authors use EA, other times they use TEA

Response: Thanks for your professional advice. We corrected it and we used the single term in the current manuscript.

(ix) “?Ì… is the relative density of the samples of about 0.36” – This is incoherent considering that

Response: Thank you very much for your thorough observation. As the section 2.3.5 (Linear elastic deflection and elastic collapse) was removed from the manuscript, this term is no longer used and subsequent topics related to it also removed from both of the Result and Discussion sections.

 (x) Table 2 shows a ~60-64% porosity. Also, standard deviations should be added.

Response: Thanks for your professional advice. Standard deviation is added to the Table 2.

(xi) “On the other hand, when the load surpasses the Euler buckling load, the cell wall of the Voronoi structures buckles, this is called the elastic collapse, which is similar to that in foam and honeycomb.” Considering that the authors statement is located in the methods section, they cannot yet prove this. In fact, even for honeycombs, this is only true for one direction. Also, honeycomb theory (especially, when citing L.Gibson and M.Ashbys work) is appropriate for high values of porosity (>90%). Considering that the samples display a porosity ~60-64% (Table 2), there are strong axial and hinging effects that affect the micromechanical deformation behavior of the samples.

Response: Thank you very much for your thorough observation. As the section 2.3.5 (Linear elastic deflection and elastic collapse) was removed from the manuscript, the subsequent topics related to it also removed from the Discussion section.

(xii) “The compressive test was repeated 4 times for each sample and the average value of these four tests was reported here” – Standard deviations should be added.

Response: Thank you very much for your thorough observation. Standard deviation is added to the Table 3.

(xiii) Lines 171 to 313 – Should be corrected to Apparent stresses, strain and moduli.

Response: Thank you very much for your thorough observation. We carefully checked the manuscript and corrected all terms of stresses, strain and moduli to Apparent stresses, strain and moduli, respectively.

(xiv) Table 3 – add standard deviations and keep the same number of decimals.

Response: Thank you very much for your professional advice. We followed your valuable suggestion.

(xv) Displacements – sometimes in “m”, others in “mm”. The same unit should be used.

Response: Thank you very much for your professional advice. We followed your valuable suggestion. As you can find from Figure 8, the dimension of the displacement was modified to mm. Also, we carefully modified it throughout the manuscript text.

Figure 1 Variation in energy absorption (EA) as a function of displacement for the six Voronoi structures. (a) homogeneous model for elongated (E1) and normal (N1) Voronoi; (b) gradient I model for elongated (E2) and normal (N2) Voronoi; (c) gradient II model for elongated (E3) and normal (N3) Voronoi.

(xvi) Fig.9 - why are so many decimals used? Keep the coherence with other tables and figures.

Response: Thank you very much for your professional advice. We followed your valuable suggestion and we used only two decimals similar to other figures and tables in this manuscript.

(xvii) Table 4 – Where the C1 and C2 values determined by testing only one topology. Usually, these factors are determined by testing the same topology with different specific densities (e.g. by changing the rib thickness) following by non-linear regression. Considering that this model is based on an exponential function, how can the authors claim that these parameters may be determined with only one result?

Response: Thank you very much for your thorough observation. As the section 2.3.5 (Linear elastic deflection and elastic collapse) was removed from the manuscript, the subsequent topics related to it also removed from the Discussion section.

(xviii) “Thus it can be concluded from this that gradient I structure design can resist bending more than the fully filled structure and therefore its value is expected to exceed 1 in the case of using a metallic material.” – considering that these models are based on normalized moduli (E*/Es), this is simply not true. Surely, the changes in C1 (admitting that these values are correct, even though they were calculated with only one point per topology) reveal changes in the deformation mechanism, however, they are not correctly interpreted.

  Response: Thank you very much for your thorough observation. As the section 2.3.5 (Linear elastic deflection and elastic collapse) was removed from the manuscript, the subsequent topics related to it also removed from the Discussion section.

(xix) "The stress-strain curve shows brittle behavior and this was expected because it is known that most polymeric materials are brittle” – This is not true, and the apparent stress-strain plots prove this. Also Reference 55 detail PLA.

Response: Thank you very much for your detailed review. Here this section was modified as following:

The failure mode is the same for all samples as illustrated in Error! Reference source not found. for homogeneous elongated Voronoi structure. The cylindrical Voronoi cells all eventually collapsed due to a failure mode involving three subsequent stages of failure starting from the fracture of the cells’ strut, cell collapse, and finally, it followed by fracture growth [31]. This demonstrates that despite the material's great strength, the samples experienced a fracture and so absorbed less energy. This is due to cracking only occurring at low strains. Notably, the densification of the structures happened at varying strains of about: 22% for homogeneous, 9% for gradient I, and 17% for gradient II structures, until the entire structure was flattened. This is made abundantly clear by the gradual rise in the strain that occurs as a result of the pores closing and the struts coming into contact with each other under higher strain [55]. A similar phenomenon of the failure was detected in the breaking behaviour of lattice structures for diverse additive manufactured materials using the same technique [56,57]. Moreover, Chao et al [58] Studied the failure nature of the Voronoi structure with different porosity levels (70, 80, and 90%.) using the Finite Element Method (FEM). They found that the maximum Mises stress (from Von Mises Yield criterion theory) from porous Voronoi structures is mainly concentrated at the nodes where the connecting rods of the porous structure are connected. They investigated that the randomly distributed pores are helpful to promote fragile and brittle pore edges which led to the concertation of more stresses. The most obvious finding in their study is the average pore size has a crucial influence on the stress distribution on the connecting rod edges. The smaller the pore size, the higher the stress concentration which leads to easier fracture. They suggested that for designing the Voronoi structure, the appropriate average cell size should be considered more than the porosity percentage. Based on the results presented in their study and combining the fact that the elongated Voronoi structures in the current study have larger pore sizes; they can carry larger apparent stresses and loading before they start to be fractured. This is a reason for the mechanical properties’ improvement in the current study.

(xx) Samples are called E1-E3 and N1-N3, however, in the conclusions section are referred as IEVs and INVs.

Response: Thank you very much for your thorough observation. These terms are used in the current study are explained in the section 2.1 as following:

Irregular Elongated Voronoi Structures (IEVSs) which in turn is classified into E1, E2 and E3 based on the homogeneous, Gradient I and Gradient II, respectively.

Irregular Normal Voronoi Structures (INVSs) which is classified into N1, N2 and N3 based on the homogeneous, Gradient I and Gradient II, respectively.  

(xxi) Final remark, considering that Voronoi structures are usually generated randomly, what is the impact of this aspect in this study?

Response: Thank you very much for your detailed review. Irregular Voronoi microstructures are better mimic the natural bone microarchitecture. Hence, they are more suitable for bone tissue growth than regular porous microstructures as staed by (Bhate et al., 2019). In the last few years, Gómez et al. (2016), Fantini and Curto (2017) and Du et al. (2020), among others, have proposed irregular porous scaffolds based on Voronoi tessellations. However, as we can see in the nature, even irregular structure developed and optimized in direction of loading, and this give them properties which depend on the direction in which they are measured. Almost most of natural cellular solid are like this, the enormous anisotropy of wood and leaf and stalk is largely caused by elongated shape of their cells. For bone scaffolds, we know the loading direction so we can propose and design the scaffold base on loading direction, that can improve mechanical properties.

By adjusting the parametric design of structures based on Voronoi tessellation, the requirements set for the mechanical properties and permeability of different porous structures are met. Currently, porous structure modeling based on Voronoi tessellation is limited by a specific modeling technique and cannot easily control the aperture and porous structure.

Reviewer 3 Report

The paper is an interesting approach to the design and 3D printing and mechanical properties of 3D Voronoi. The authors attempted to design some sample inspiring from the nature and determine its mechanical properties. The paper organization and arrangement are fine and acceptable. It is will be useful for reader to add more literature on 3D Voronoi and its different fabrication methods in introduction part.

Comments

1-    There are mistypes in the paper such as the title for graphical abstract. Please correct it and check the entire of paper for other mistypes as well.

2-    When an abbreviation is defined in a text, the abbreviation should be used on the rest of the paper. Otherwise, defining an abbreviation has no sense. For example, tissue engineering (TE), is defined in line 26, but is not used in the next sentence and other places. Please check all abbreviations for these small mistakes. Another example is “EA” which defined but not used.

3-    Please revise the sentence in line 100 of the PDF.

4-    The figure 1 mistakenly repeated in page 3 of the PDF. Its quality also should be improved. Or use a new figure with higher quality.

5-    Quality of the graph and images should be improved

6-    Strain rate during the compressive testing should be mentioned in the methodology part.

7-    It would be useful to add the grasshopper diagram as appendix of paper,

The revised paper will be accepted upon considering the above comments.

Author Response

Dear Reviewer,

Thank you very much for your time involved in reviewing the manuscript and your very encouraging comments on the merits.

Comments and Suggestions for Authors

The paper is an interesting approach to the design and 3D printing and mechanical properties of 3D Voronoi. The authors attempted to design some sample inspiring from the nature and determine its mechanical properties. The paper organization and arrangement are fine and acceptable. It is will be useful for reader to add more literature on 3D Voronoi and its different fabrication methods in introduction part.

Response: Thank you for your professional advice. We will consider your valuable recommendations.

Comments

1-    There are mistypes in the paper such as the title for graphical abstract. Please correct it and check the entire of paper for other mistypes as well.

Response: Thank you for your professional advice. We corrected this spelling mistake and corrected the whole spelling mistakes throughout the manuscript.

2-    When an abbreviation is defined in a text, the abbreviation should be used on the rest of the paper. Otherwise, defining an abbreviation has no sense. For example, tissue engineering (TE), is defined in line 26, but is not used in the next sentence and other places. Please check all abbreviations for these small mistakes. Another example is “EA” which defined but not used.

Response: We really appreciate your suggestions. We followed your valuable suggestions, and all abbreviations are written based on the first definition throughout the text.

3-    Please revise the sentence in line 100 of the PDF.

       Response: Thank you very for your thorough observation. We appreciate your time and effort to correct our manuscript to meet the standard requirement for publication in high reputed journals. We corrected this typo error.

4-    The figure 1 mistakenly repeated in page 3 of the PDF. Its quality also should be improved. Or use a new figure with higher quality.

Response: Thank you for your professional advice. We removed the repeated Figure 1, and its quality is improved.

5-    Quality of the graph and images should be improved.

Response: Thank you for your professional advice. The quality of figures was improved.

6-    Strain rate during the compressive testing should be mentioned in the methodology part.

Response: Thank you very much for your thorough observation. This phrase was added in the methodology section:

The test was done in quasi static condition at strain rate of 0.001/sec.

7-    It would be useful to add the grasshopper diagram as appendix of paper.

Response: Thank you very much for your valuable observation. We added grasshopper diagrams as appendix of the paper.

The revised paper will be accepted upon considering the above comments.

Round 2

Reviewer 2 Report

(i) On the stress-strain curves: “This is due to cracking only occurring at low strains. Notably, the densification of the structures happened at varying strains of about: 22% for homogeneous, 9% for gradient I, and 17% for gradient II structures, until the entire structure was flattened.” – How can densification strains be discussed if this data is not shown? Figures 6 and 7 do not show the densification stage as it would be expected. Also, it is widely known that the densification strain is highly dependent on the sample porosity (approximately the same value as the porosity, as this is the empty space in which the sample can deform before densifying). How did the authors determine that the densification occurs at such lo values? The authors themselves claim that “This is made abundantly clear by the gradual rise in the strain that occurs as a result of the pores closing and the struts coming into contact with each other under higher strain”.

(ii) “Cosma et al. [31] reported three failure modes for scaffold based on lattice structure with 56% porosity, which they are buckling, bending” – Authors should be aware that the word buckling cannot be fount in reference [31], it simply gives no support to this statement.

(iii) Considering that Voronoi structures are usually generated randomly, what is the impact of this aspect in this study? The authors did not really clarify this issue in the revision. If the topology for the E and N samples are generated randomly, how can the authors expect, that if this protocol is repeated, the results are similar? For instance, if I take the conditions of sample N1 and repeat the seeding and modeling protocols on grasshopper, two different topologies will be generated! How can the authors assure with certainty that all the claims in the manuscript are true?

(iv) The authors should take their time to prepare the manuscript and its revision. There are certain aspects that are clearly neglected and reveal that the manuscript is not really ready to be submitted (e.g. Figures in Appendix A cannot be read and some of them are cut; Revision with Error! Reference source not found issues).

Author Response

Dear Reviewer,

Thank you very much for your time involved in reviewing the manuscript and your very encouraging comments on the merits.

Comments and Suggestions for Authors

The manuscript shows details a novel topology, referred as Elongated Voronoi structures and compares their behavior to “normal” Voronoi Structures. The authors produce variations of these models by SLA and test them their mechanical behavior by uniaxial compression. Conclusions are drawn to compare their properties and establish the benefits of this new model. There are, however, numerous shortcomings that compromise the quality of the manuscript:

Response: Thank you for your professional advice. We will consider your valuable recommendations. They are helpful to improve the quality of the current manuscript to make it satisfactory for the publication in high reputed journals.

Note:
1- previous revision was highlighted in yellow. Second round revisions are highlighted in blue the second-round revision of the current manuscript.

2- Despite the response to the valuable comments of the reviewers, we assigned English editing and corrected several aspects of the manuscript. These new corrections are highlighted in blue.

  • On the stress-strain curves: “This is due to cracking only occurring at low strains. Notably, the densification of the structures happened at varying strains of about: 22% for homogeneous, 9% for gradient I, and 17% for gradient II structures, until the entire structure was flattened.” – How can densification strains be discussed if this data is not shown? Figures 6 and 7 do not show the densification stage as it would be expected. Also, it is widely known that the densification strain is highly dependent on the sample porosity (approximately the same value as the porosity, as this is the empty space in which the sample can deform before densifying). How did the authors determine that the densification occurs at such lo values? The authors themselves claim that “This is made abundantly clear by the gradual rise in the strain that occurs as a result of the pores closing and the struts coming into contact with each other under higher strain”.

Response: Thank you for your professional advice. We removed this part from the manuscript. We agree with your comment.

This explanation was based on the previous stress-strain curved which they were plotted based on the Hook’s law. Thankfully, the reviewer advised us to redraw these curves for non-linear behaviours. We redrew these curves, but we forgot to remove this explanation in the revised manuscript. We strongly apologize for this. Thank you for your careful observation and guidance to make our manuscript more professional.

  • “Cosma et al. [31] reported three failure modes for scaffold based on lattice structure with 56% porosity, which they are buckling, bending” – Authors should be aware that the word buckling cannot be fount in reference [31], it simply gives no support to this statement.

Response: Thank you for your professional advice. We apologize for this mistake. We removed this reference from the manuscript.

  • Considering that Voronoi structures are usually generated randomly, what is the impact of this aspect in this study? The authors did not really clarify this issue in the revision. If the topology for the E and N samples are generated randomly, how can the authors expect, that if this protocol is repeated, the results are similar? For instance, if I take the conditions of sample N1 and repeat the seeding and modeling protocols on grasshopper, two different topologies will be generated! How can the authors assure with certainty that all the claims in the manuscript are true?

Response:  Thank you for your thorough observation. Our main focus is the study of effect cell type in Voronoi. Especially, when we elongated it on the loading direction for possible application as bone implant, so we kept the porosity constant. for sure further investigation is necessary about this topic to provide better understanding of cell geometry effect However, in this research we limit our investigation on 6 proposed samples and their mechanical properties, and we hope this research might be starting point in this topic.

  • The authors should take their time to prepare the manuscript and its revision. There are certain aspects that are clearly neglected and reveal that the manuscript is not really ready to be submitted (e.g. Figures in Appendix A cannot be read and some of them are cut; Revision with Error! Reference source not found issues).

Response: Thank you for your professional advice. We agree with your comment.

We modified the figures in the Appendix A. We added an explanation in the Appendix A and embedded more clear figures into the manuscript. The references are also updated.
